# Structure and Dielectric Properties of TPU Composite Filled with CNTs@PDA Nanofibers and MXene Nanosheets

**DOI:** 10.3390/polym14112157

**Published:** 2022-05-26

**Authors:** Zhaoxia Luo, Xiaolin Li, Suhe Zhao, Lianghua Xu, Li Liu

**Affiliations:** 1State Key Laboratory of Chemical Resource Engineering, Beijing University of Chemical Technology, Beijing 100029, China; sunglow11@sina.com (Z.L.); lixl@mail.buct.edu.cn (X.L.); zhaosh@mail.buct.edu.cn (S.Z.); 2Beijing Engineering Research Center of Advanced Elastomers, Beijing University of Chemical Technology, Beijing 100029, China; 3National Carbon Fiber Engineering Research Center, Beijing University of Chemical Technology, Beijing 100029, China; 4Carbon Fiber and Functional Polymer Key Laboratory of Ministry of Education, Beijing 100029, China

**Keywords:** dielectric elastomer, CNTs, PDA, MXene

## Abstract

Thermoplastic polyurethane (TPU) is a kind of dielectric elastomer (DE) which can behave as an actuator, altering thickness strain in response to electrical stimulation. The composites are made up of fillers with a very high dielectric constant that are spread in a polymer matrix. It is very difficult to obtain large deformation at low voltage. In this study, we made two-dimensional (2D) MXene nanosheets with excellent conductivity and one-dimensional (1D) polydopamine (PDA)-modified CNT fiber fillers. After that, TPU dielectric elastomer films made of MXene/CNTs or MXene/CNTs@PDA were prepared. The results showed that the dielectric constant and dielectric loss of TPU dielectric film including MXene/CNTs were much higher than that containing MXene/CNTs@PDA, although Young’s modulus and breakdown strength (*E_b_*) were significantly lower. At the same time, these two types of dielectric films had a significantly higher dielectric constant and dielectric loss than pure TPU dielectric film, and their breakdown strength was significantly lower. The compatibility of CNTs@PDA fibers with the TPU matrix improves after PDA modification, and the dispersion of CNTs@PDA fibers improves, resulting in an increase in Young’s modulus. MXene with a two-dimensional nanosheet structure increases the breakdown strength of the TPU dielectric elastomer under the condition of the addition of a tiny quantity. To summarize, the dielectric constant, dielectric loss, Young’s modulus, and dielectric elastomer breakdown strength are mutually restrictive conditions, and the relationship between all parties must be balanced to obtain obvious deformation properties.

## 1. Introduction

A dielectric elastomer is a smart material capable of electrically induced deformation [1,2]. Under the influence of an external electric field, it may compress vertically and expand horizontally. It has a wide range of applications in intelligent prosthesis [3,4], electronic skin [5], artificial muscles [6,7,8,9,10], and other fields because of its unique ability to deform vast amounts of material [11]. Researchers have performed many scientific studies on this topic in recent years [9,12].

However, dielectric elastomers can only deform significantly under the action of high voltage, and high voltage (>70 kV/mm) poses a great threat to human safety [13,14]. Therefore, obtaining large deformation at low voltage is one of the technical difficulties encountered by dielectric elastomers [15,16]. The strain of DE could be attributed to the Maxwell stress effect [12], and most of the previous research is focused on how to enhance the dielectric constant or reduce Young’s modulus [17,18].

Carbon nanotubes (CNTs) are considered the most promising nanomaterials because of their great flexibility, low mass density, high aspect ratio (usually greater than 10^3^), and excellent electrical properties [19,20,21]. However, ordinary CNTs are difficult to evenly disperse in the rubber matrix and are unable to form a good bonding interface with rubber. Therefore, a modification method is used to promote its combination with the rubber matrix [22,23]. MXenes, a new family of 2D transition metal carbides and carbonitrides, has outstanding conductivity and a hydrophilic surface; therefore, the addition of a small amount can significantly improve the dielectric constant without increasing Young’s modulus [24,25,26].

Despite the advantages of high conductivity of one-dimensional (1D) CNTs@PDA fibers and two-dimensional (2D) MXene nanosheets, there is no research reported on the combination of these two different dimensions (1D/2D) of conductive fillers in the TPU matrix.

One-dimensional CNT highly conductive fillers were employed in this investigation, based on our earlier work. To make CNTs compatible with TPU, the surface was modified with polydopamine [27]. We combined the structural and high conductivity characteristics of these two conductive fillers (1D CNTs@PDA and 2D MXene) under the condition that a small amount of filling can quite improve the dielectric constant of the dielectric elastomer. By not adding a considerable amount of high dielectric filler, it should not result in a significant rise in the modulus. The codoped high-conductivity MXene two-dimensional lamellar structure material raises the permittivity of the composites much further. It is desirable to obtain a dielectric elastomer with a large deformation.

## 2. Experimental Section

### 2.1. Materials

Titanium aluminum carbide (MAX) was obtained from Jilin Yiyi Technology Co., Ltd., Changchun, China. Dopamine and trihydroxymethylaminomethane both were purchased from Qingdao J&K Co., Ltd., Qingdao, China. Polyvinylpyrrolidone (PVP) was supplied by Sigma-Aldrich (St. Louis, MO, USA). Deionized (DI) water was homemade. TPU (Soft 35A) was obtained from Shanghai BASF Co., Ltd., Shanghai, China. Dimethyl sulfoxide (DMSO) and lithium fluoride (LiF) were purchased from Aladdin Technology Co., Ltd. (Shanghai, China). CNTs were purchased from Beijing Cnano Technology Co., Ltd., Beijing, China. Anhydrous ethanol and concentrated hydrochloric acid (HCl) were supplied by the Beijing Chemical Plant, Beijing, China.

### 2.2. The Preparation of Sample

#### 2.2.1. The Preparation of Ti_3_C_2_T_x_ (MXene)

An amount of 1 g of LiF was dissolved in 20 mL of HCl (9 M) and magnetically stirred for 30 min, then 1 g of MAX was progressively added to the aforesaid solution in a water-bath at 35 °C and magnetically stirred for 24 h. The etched MAX solution was placed into a centrifuge tube, spun, and rinsed many times. Finally, the etched MAX was vacuum dried to produce MXene [26].

#### 2.2.2. Modification of CNT Nanofibers

In 50 mL of DI water, 0.06 g of trihydroxymethylaminomethane was dissolved. By using a pH meter, the pH was adjusted to 8.5 with 1 M of HCl, and 0.1 g of dopamine hydrochloride was added to the above-mentioned solution and mixed uniformly. The mixed solution was magnetically stirred for 24 h with a suitable number of CNT nanofibers. The CNT fiber coated with PDA was then washed with DI water several times and dried at room temperature to obtain CNTs@PDA nanofibers.

#### 2.2.3. The Preparation of TPU Solution

To make a 25 wt% solution, TPU granules were dissolved in DMSO solvent. The flask was capped and heated until the TPU particles were completely and equally dissolved in DMSO in an air dry oven at 70 °C.

#### 2.2.4. The Preparation of MXene/CNTs@PDA/TPU Composite

Ultrasonication was used to mix an appropriate amount of CNTs@PDA nanofibers with MXene solution for 30 min. The blended solution was then ultrasonic for 2 h after the combination was put into the prepared TPU solution. Afterwards, the solution was poured into a plastic mold, and the resulting samples were dried at 60 °C to eliminate the solvents. Finally, the different filler content samples were collected.

### 2.3. Characterization

The morphologies of the CNT nanofibers and TPU nanocomposites were studied using scanning electron microscopy (SEM, S-4800, Hitachi Co., Tokyo, Japan). The elemental mapping images of the various elements present in the MAX were described using energy dispersive spectroscopy (EDS). Using an HRTEM Titan H-9500 apparatus, high-resolution transmission electron microscope pictures were obtained. A Shimadzu XRD-6000 diffractometer was used to obtain an X-ray diffraction (XRD) pattern (Cu-Kα radiation λ = 0.154 nm). The chemical structure of MXene was determined using a Raman spectrometer (Invia Reflex, Renishaw, UK). Atomic force microscopy (AFM) was utilized to investigate morphological characteristics, and the height maps were employed to characterize the thickness of MXene. An Escalab 250 XPS system (Thermo Electron Corporation, Waltham, MA, USA) with an Al-Kα X-ray source was used to gather X-ray photoelectron (XPS) data. Tensile testing equipment with a testing speed of 500 mm/min was used to obtain stress–strain curves. The specimens used were dumbbell-shaped with a width of 4 mm and a thickness of about 0.5 mm. When the strain was 5%, the slope value of the stress–strain curve was used to compute Young’s modulus. Furthermore, a broadband Novocontrol dielectric spectrometer equipped with an Alpha-A frequency analyzer (Germany) was also used to determine the dielectric characteristics in the frequency range of 10^0^ to 10^7^ Hz at room temperature. The dielectric constant and dielectric loss of the samples were measured twice at different positions. Conductive carbon black, silicone oil, and solvent were used to make the electrode liquid. Two circular acrylic glass molds with 10 mm holes were clipped to both sides of the dielectric film, and then an electric spray gun was used to spray the electrode liquid over both sides of the elastomer film. After that, the electrode liquid-coated sample was connected to the high-voltage power source using the high-voltage power supply. The test value for breakdown strength was the ratio of the greatest voltage to the sample thickness when the sample was broken down. The test conditions were 25 °C and 30% humidity.

## 3. Results and Discussion

### 3.1. The Characterization of MAX and MXene

The EDS spectrum of MAX, which is mostly constituted of titanium, carbon, aluminum, and oxygen, is depicted in Figure 1a–d. From the mapping images, the transverse size distribution of the MAX lamella, which was concentrated around 4–18 μm and was mostly in the 4–10 μm range, was tallied.

Figure 2a shows the raw material photos of titanium aluminum carbide (MAX), which show that it was black powder. As shown in Figure 2b,c, the raw MAX has a flat surface and a high thickness. After the Al components were etched, the thickness of the different layers in Figure 2d was about 50 nm. After a long duration of ultrasonic peeling, vacuum filtering produced a thin layer with the lamella structure shown in Figure 2e. Figure 2f illustrates a sheet layer structure of MXene, revealing that a few layers of MXene were effectively produced after etching Al atom layers from the MAX phase using ultrasonic liquid-phase stripping.

The MAX peaks at 9.5° (002) and 38.7° (008) are seen in Figure 3a. After HCl etching and sonication, the peak in the XRD spectrum that was initially found at 9.5° changed to 7.5°, showing that the interlayer spacing was increased from 9.4 to 11.8 Ȧ. MXene displayed a sharp (002) peak, indicating that it has a highly organized crystalline structure following etching. The strong peak of the Al atomic layer at 38.7° was clearly attenuated, showing that MXene was produced by etching MAX. The Raman spectrum in Figure 3b exhibits MXene-specific peaks at 203.7 cm^−1^. The Raman spectrum in Figure 3b reveals distinctive MXene peaks at 203.7 cm^−1^ and 722 cm^−1^, showing that MXene was successfully prepared. An atomic force microscope (AFM) was used to investigate the few-layer MXene in Figure 3c. The peeled MXene had a thickness of about 2–4 nm, indicating that there were only a few layers. The surface chemical properties of MAX and MXene were investigated using X-ray photoelectron spectroscopy (XPS), as illustrated in Figure 3d–f. In addition, the spectra showed no trace of an Al 2p peak, indicating that the Al atomic layer in the MAX was erased. In Figure 3d, Ti 2p, O 1s, C 1s, and F 1s peaks with binding energies of 456, 531, 285, and 685 eV are clearly visible, respectively.

### 3.2. The Modification of CNTs

The structure and surface features of carbon nanotubes (CNTs) and carbon nanotubes modified by polydopamine (CNTs@PDA) were investigated in this work, as illustrated in Figure 4. The distinctive peak of D, G, and G’ in the Raman spectra of CNTs (see Figure 4a) was 1350 cm^−1^, 1580 cm^−1^, and 2700 cm^−1^, respectively. The G band at 1584 cm^−1^ had a greater peak than the peak at 1355 cm^−1^. The occurrence of lattice distortion or structural flaws, which was generated by chemical groups at various places formed during the manufacture of CNTs, is related to the disorder-induced D band.

The amorphous carbon peak of CNTs emerged around 20–30 degrees, according to the XRD spectrum of CNTs (see Figure 4b). Following surface biomimetic modification by polydopamine, the amorphous peak of CNTs shifted to the right, and the overall peak shape changed slightly, indicating that the crystal structure of CNTs was not altered after polydopamine was coated on the surface of CNTs.

The CNTs had a diameter of 30–40 nm and a length of 1–5 nm, as shown in Figure 4c,d. The surface chemical properties of CNTs and CNTs@PDA were analyzed by XPS, as shown in Figure 4e,f, clearly showing the presence of O 1s, C 1s, and N 1s peaks at 531, 285, and 400 eV of binding energy. The existence of the N1s peak indicates that polydopamine was coated on the surface of the CNTs.

### 3.3. Micro-Morphology of TPU Composites

The cross-sectional SEM photos of TPU composites codoped with different fillers are showed in Figure 5. Figure 5a shows that the cross-sectional SEM images of pure TPU were free of impurities and the surface was smooth. In Figure 5b, the addition of MXene led to a rough cross-section and many folds in the compound, and no obvious two-dimensional lamellar material agglomeration was found, indicating that oxygen, fluorine, hydroxyl, and other groups on the surface of MXene can interact with nitrogen and oxygen on the TPU molecular chain, improving their compatibility, strengthening the interface, and promoting dispersion at the same time. Figure 5c,d was the cross-sectional SEM images of TPU composites with different amounts of CNTs. It can be seen from the figure that CNTs were distributed in the TPU matrix in the form of a short rod. With the increase of CNT content, the dispersion of CNTs in TPU was still uniform and there was no obvious agglomeration. The results showed that CNTs can be uniformly dispersed in a TPU matrix under the action of a liquid-phase ultrasound for a long time.

In Figure 5e,f, the TPU matrix was filled with the polydopamine (PDA)-modified CNTs. Due to the interface between PDA and TPU molecules, the dispersion of CNTs was strengthened, and it was difficult to observe the CNT particles. Figure 5g,h shows the SEM cross-sectional microphotos of TPU composites codoped with CNTs and MXene fillers. It can be seen from the figure that there were no obvious agglomerations in the TPU matrix under the condition of different codoped fillers, and CNTs were disorderly arranged in the composite. When the CNT filler fraction increased to 0.6 wt%, the number of carbon nanotubes observed in SEM images increased significantly. In Figure 5i,j, MXene and CNTs@PDA mixed together and added to the TPU matrix, and MXene and CNTs@PDA evenly dispersed, with no aggregation, in a disordered state. Therefore, when the amount of filler increases, the amount of filler observed in SEM images increases significantly.

Clearer particle dispersion can be observed by a high-resolution transmission electron microscope (HRTEM) (see Figure 6). When the quantity of CNTs and MXene were relatively small, it was difficult to see MXene particles in the images, and only a small number of CNTs were distributed in the TPU matrix (see Figure 6a,b). When the content of CNTs and MXene increased, as shown in Figure 6c,d, more gray tubular CNT particles can be observed, and winding aggregation occurred in local areas. Some black massive particles were also distributed. Few-layer MXene also had aggregation. These microstructure differences also determine the dielectric properties and breakdown strength of the codoped system.

### 3.4. The Mechanical Properties of Dielectric Composites 

TPU dielectric composites codoped with MXene and CNT particles showed excellent mechanical properties, as shown in Figure 7a. The stress and strain properties of 0.8 wt% CNTs/1 wt% MXene/TPU composites are equivalent to those of pure TPU, but the modulus is significantly improved. Although MXene was introduced into the CNTs/TPU composite because its surface oxygen, fluorine, and hydroxyl groups acted with –NH– and C=O on TPU, the increase in elongation at the break caused by the increase in molecular movement ability of soft segments was conducive to the orientation of TPU molecules on the MXene surface, TPU matrix soft segment molecules, and CNT particles along the long axis of tensile direction; thus, it played a good role in orientation enhancement.

TPU dielectric composites codoped with MXene and CNTs@PDA particles showed similar mechanical properties, as shown in Figure 7b. Young’s modulus of the TPU composites increased with the increase in the amount of codoped filler.

Hysteresis loss is an essential criterion for evaluating DEs. Figure 8 shows the hysteresis loss of MXene/CNTs@PDA/TPU dielectric composites. It was found that the hysteresis loss of the TPU composite increased as the MXene and CNTs@PDA content grew, yet the hysteresis loss was lower than pure TPU. The reason for this is that the more filler content there is, the more friction there is between the TPU molecular chain and filler particles, which is prone to loss.

### 3.5. The Dielectric Properties of TPU Composites with MXene/CNTs or MXene/CNTs@PDA

Figure 9 shows the frequency dependence of dielectric constants and losses in TPU composites. The dielectric constant and loss factor of codoped CNTs/MXene and CNTs@PDA/MXene TPU composites were both high. The dielectric constant of the composite increased gradually as the amount of codoped filler increased in Figure 9a–d, and the dielectric loss likewise increased gradually. When the electric field frequency was at 10^3^ Hz, the dielectric constant of the TPU composite codoped with 0.5 wt% MXene and 0.6 wt% CNTs was 26.11 and the dielectric loss was 0.463, whereas the dielectric constant of the TPU composite codoped with 0.5 wt% MXene and 0.6 wt% CNTs@PDA was 12.5 and the dielectric loss was 0.382.

Although the dielectric constant of these TPU composites was higher than a pure TPU matrix, the dielectric loss remained relatively unchanged. This is mostly owing to the use of high conductivity fillers such as MXene and carbon nanotubes. 

At the same time, due to the functional groups on the surface of MXene that can react with the functional groups on the PDA molecular chain of the CNTs@PDA fiber, the dielectric constant of TPU dielectric composites codoped with CNTs@PDA and MXene filler was significantly higher than that of the TPU composites codoped with CNTs or MXene filler, and the dielectric loss was reduced. According to the findings (see Figure 9e,f), the presence of a synergistic effect boosts the dielectric characteristics of TPU composites with codoped fillers. 

The results showed that when an electric field was given to the TPU composite codoped with CNTs/MXene, the microelectric capacitor formed by MXene was filled with CNT particles, and the induced charge on the surface of the CNTs flowed to the surface of the MXene rather than the flexible electrode surface. Because the plane conductivity of the MXene is low, charge conduction to the flexible electrode is limited. The vertical leakage current impact is minimized, and the huge rise in dielectric loss is effectively regulated (see Figure 10 for the schematic diagram of the microcapacitor and Figure 11 for the schematic diagram of the interface reaction).

As can be seen in Table 1, the TPU composite with 0.25 wt% MXene and 0.4 wt% CNTs had a high electrical sensitivity factor (15.32) at 1 kHz, but a low breakdown strength (2.44 kV/mm). The composite with 0.25 wt% MXene and 0.4 wt% CNTs@PDA had an electrical sensitivity factor (1.02) at 1 kHz, but a high breakdown strength (21.09 kV/mm). The TPU composite with 0.5 wt% MXene and 0.6 wt% CNTs had a high electrical sensitivity factor (12.67) at 1 kHz, but a low breakdown strength (0.397 kV/mm). The composite with 0.5 wt% MXene and 0.6 wt% CNTs@PDA had a low electrical sensitivity factor (1.08) at 1 kHz, but a high breakdown strength (5.04 kV/mm). The electrical sensitivity factor is inversely related to the breakdown strength for the TPU dielectric elastomer with codoped filler, as seen in the above data. The TPU material with CNTs/MXene has a high dielectric constant and a big electrical sensitivity factor, but it has a poor breakdown strength. The TPU material with CNTs@PDA/MXene has a low dielectric constant and a low power sensitivity factor, but it has a higher breakdown strength.

### 3.6. The Interface between Filler Particles and TPU Matrix

Because the interface has a considerable influence on the physical and chemical characteristics of materials, the filler is incorporated into the polymer matrix to generate a broad interface region, which plays a vital role in the performances of nanocomposites. Therefore, the modulus of the MXene/CNTs/TPU composite was studied. The atomic force microscope (AFM) logarithmic modulus diagram of MXene/CNTs/TPU composites filled with different contents are shown in Figure 12. As seen by the AFM map, when the TPU composites codoped with 0.5 wt% MXene and 0.6 wt% CNTs or 1 wt% MXene and 0.8 wt% CNTs, the inorganic particles in TPU composites were agglomerated. The TPU dielectric elastomer codoped with MXene and CNTs@PDA fillers had the same interface properties.

## 4. Conclusions

In this study, CNT nanofibers, PDA-modified CNT nanofibers, and MXene (Ti_3_C_2_T_x_) were added to a TPU matrix using a solution casting method to prepare CNTs/TPU, CNTs@PDA/TPU, MXene/CNTs/TPU, and MXene/CNTs@PDA/TPU composite materials. The findings reveal that with minor strain, a tiny quantity of high-conductivity MXene and high-dielectric constant CNTs or CNTs@PDA nanofibers may prepare double-doped filler network structures without increasing Young’s modulus. It shows that adding PDA-modified CNT filler increases the breakdown strength while lowering the dielectric constant. Therefore, in order to obtain a large strain, it should strike a compromise between these two factors, improving the dielectric constant and reducing the modulus as much as possible while assuring an increase in breakdown strength. The major reasons for enhancing the dielectric constant of the doped system are the microcapacitor structure and interfacial polarization caused by MXene nanosheets, as well as the interfacial polarization caused by CNTs and CNTs@PDA nanofibers. The laminated structure of MXene nanosheets may also inhibit the conduction of the leakage current and obtain a suitable high breakdown strength. This research may directly influence the development of low-modulus flexible TPU dielectric composite.

## Figures and Tables

**Figure 1 polymers-14-02157-f001:**
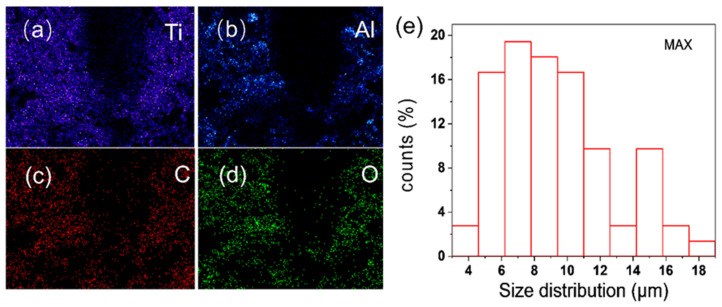
(**a**–**d**) SEM mapping map of titanium aluminum carbide (MAX). (**e**) The size distribution of MAX.

**Figure 2 polymers-14-02157-f002:**
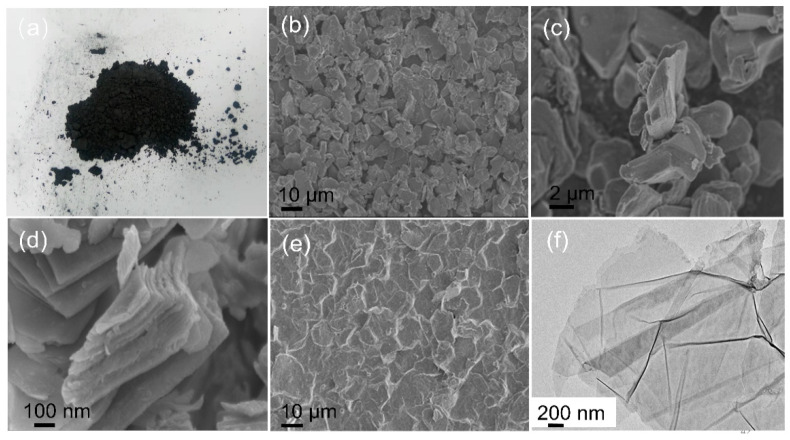
(**a**) Image of Ti_3_AlC_2_ (MAX). (**b**,**c**) SEM images of MAX. (**d**) SEM images of multi-layer MXene (Ti_3_C_2_T_x_) and (**e**) vacuum-filtered MXene. (**f**) HRTEM images of MXene.

**Figure 3 polymers-14-02157-f003:**
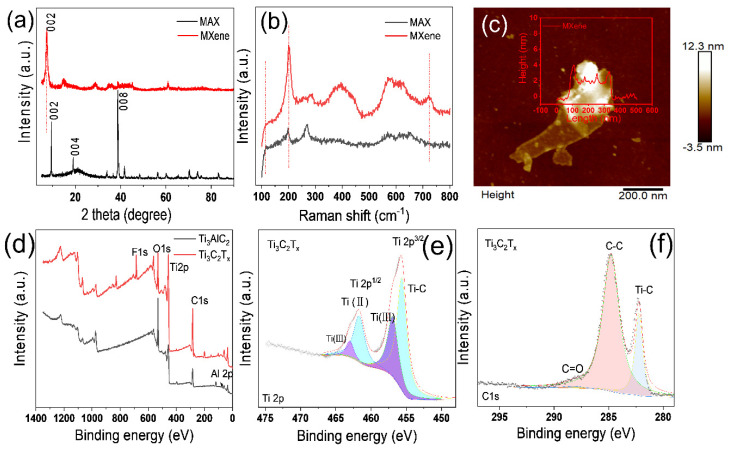
(**a**) A comparison XRD patterns of MAX phase (Ti_3_AlC_2_) and MXene (Ti_3_C_2_T_x_). (**b**) A comparison Raman spectra of MAX phase and MXene. (**c**) MXene photos from AFM microscope. (**d**) XPS spectrum of MAX phase and MXene. (**e**,**f**) XPS spectrum of Ti 2p, C1s of MXene.

**Figure 4 polymers-14-02157-f004:**
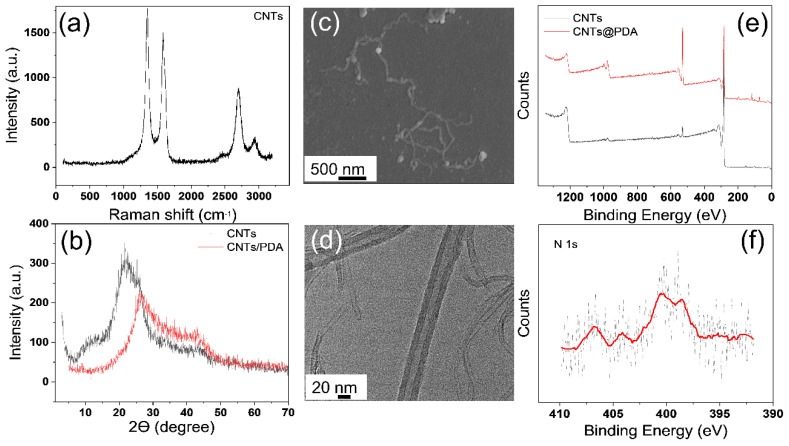
(**a**) The Raman spectrum of CNTs, (**b**) a comparison of XRD patterns of CNTs and CNTs@PDA, (**c**) SEM image of CNTs, (**d**) HRTEM image of CNTs@PDA, (**e**) a comparison of XPS spectra of CNTs and CNTs@PDA, (**f**) XPS spectrum of N1s of CNTs@PDA.

**Figure 5 polymers-14-02157-f005:**
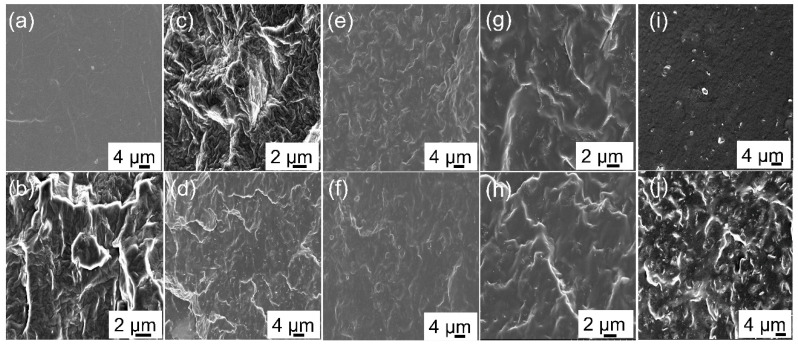
SEM cross-section images of TPU composites with various fillers. (**a**) Pure TPU, (**b**) 0.5 wt% MXene, (**c**) 0.4 wt% CNTs, (**d**) 0.6 wt% CNTs, (**e**) 0.4 wt% CNTs@PDA, (**f**) 0.6 wt% CNTs@PDA, (**g**) 0.25 wt% MXene/0.4 wt% CNTs, (**h**) 0.5 wt% MXene/0.6 wt% CNTs, (**i**) 0.25 wt% MXene/0.4 wt% CNTs@PDA, (**j**) 0.5 wt% MXene/0.6 wt% CNTs@PDA.

**Figure 6 polymers-14-02157-f006:**
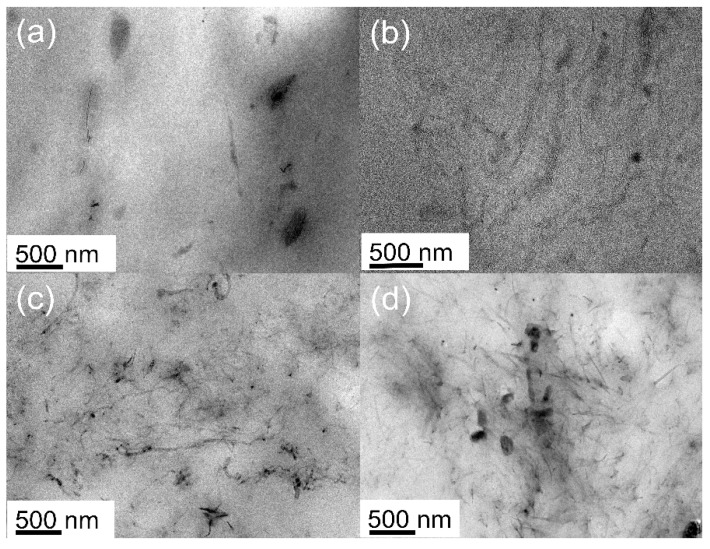
HRTEM images of CNTs/MXene/TPU composite: (**a**) 0.1 wt% MXene/0.2 wt% CNTs, (**b**) 0.25 wt% MXene/0.4 wt% CNTs, (**c**) 0.5 wt% MXene/0.6 wt% CNTs, (**d**) 1 wt% MXene/0.8 wt% CNTs.

**Figure 7 polymers-14-02157-f007:**
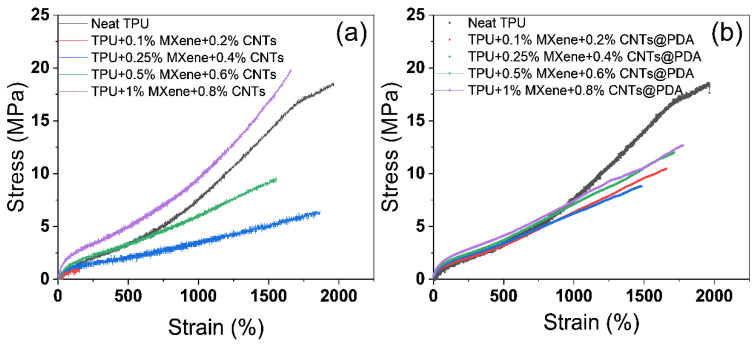
Stress–strain curves for TPU composites containing different fillers. (**a**) MXene/CNTs, (**b**) MXene/CNTs@PDA.

**Figure 8 polymers-14-02157-f008:**
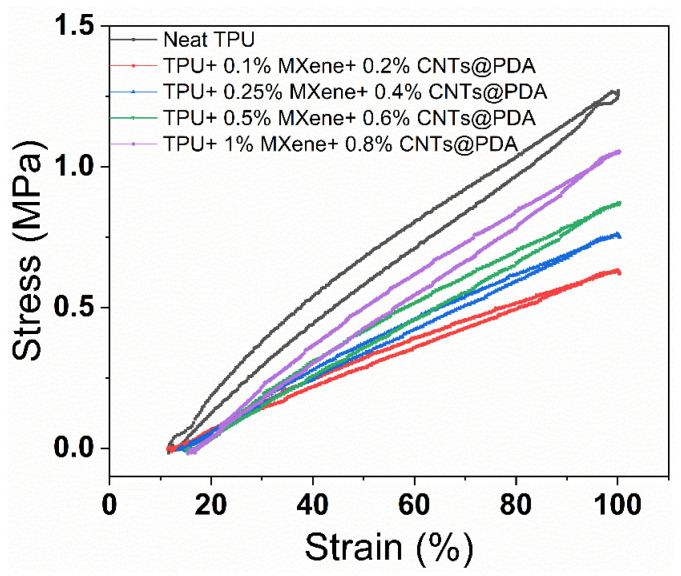
The hysteresis loss properties of TPU composites codoped with MXene and CNTs@PDA.

**Figure 9 polymers-14-02157-f009:**
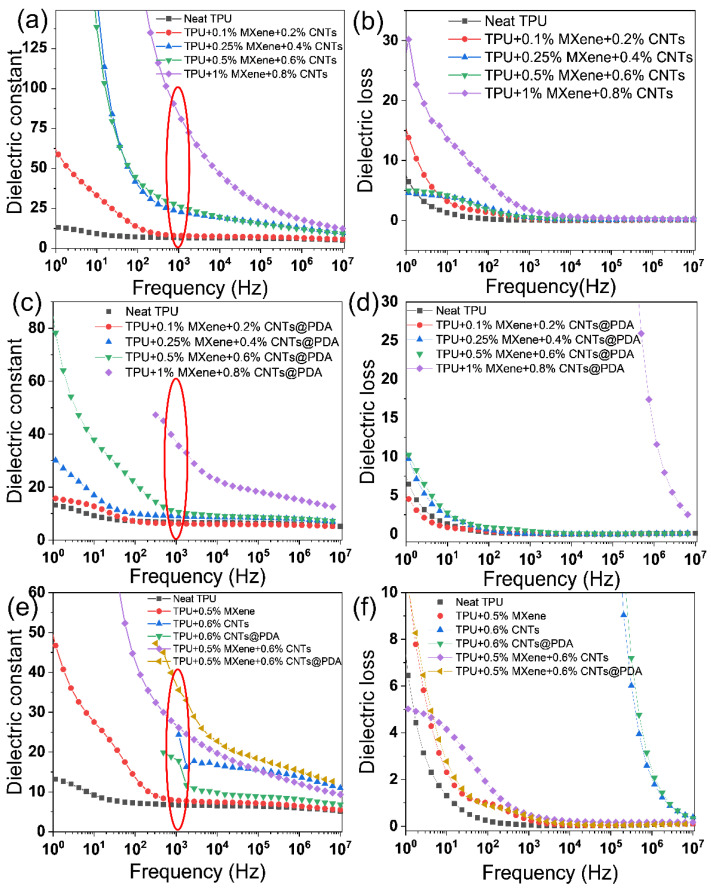
(**a**) Dielectric constant and (**b**) dielectric loss of MXene/CNTs/TPU composite with different filler contents. (**c**) Dielectric constant and (**d**) dielectric loss of MXene/CNTs@PDA/TPU composite films with various filler contents. (**e**) Dielectric constant and (**f**) dielectric loss of TPU composite films with various filler contents.

**Figure 10 polymers-14-02157-f010:**
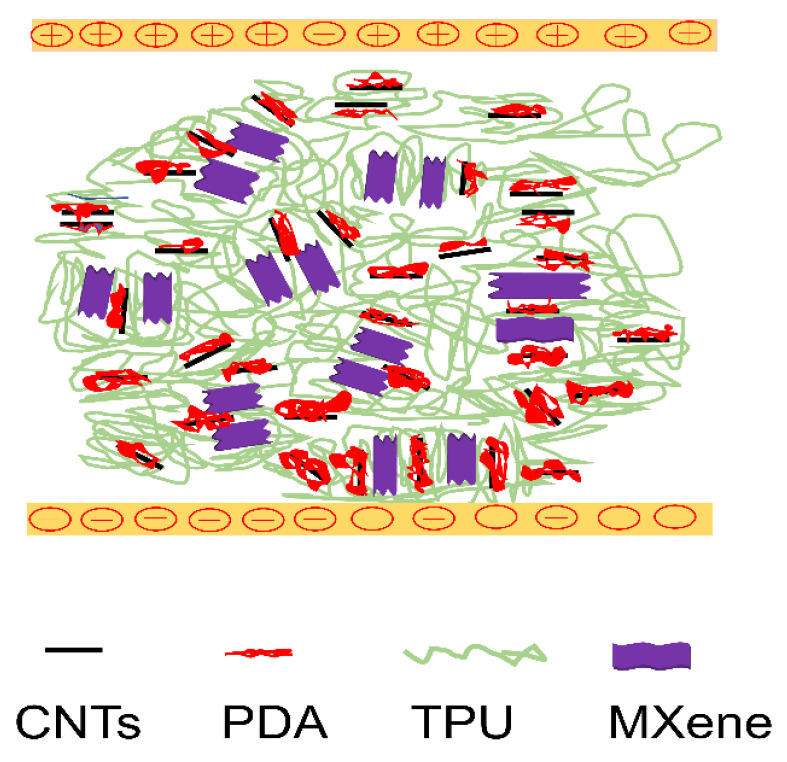
Schematic diagram of CNTs/MXene/TPU composite microcapacitor.

**Figure 11 polymers-14-02157-f011:**
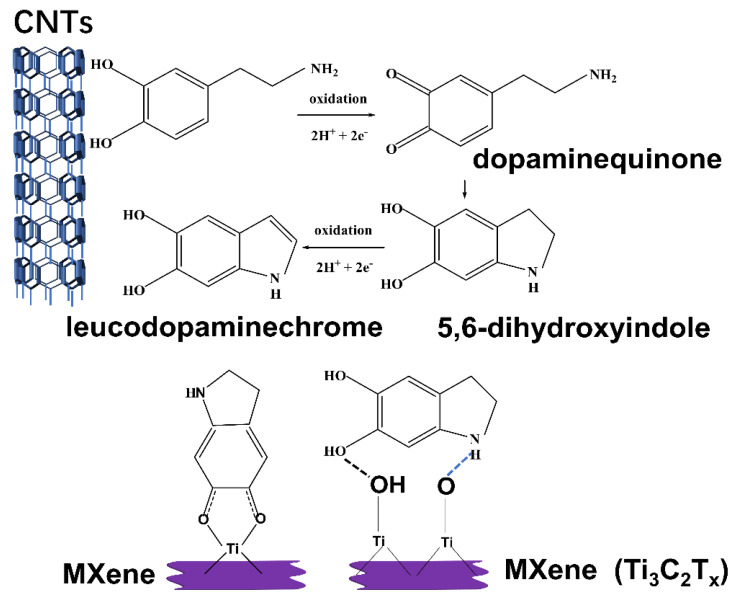
Schematic diagram of surface functional group reaction of CNTs/MXene/TPU composite.

**Figure 12 polymers-14-02157-f012:**
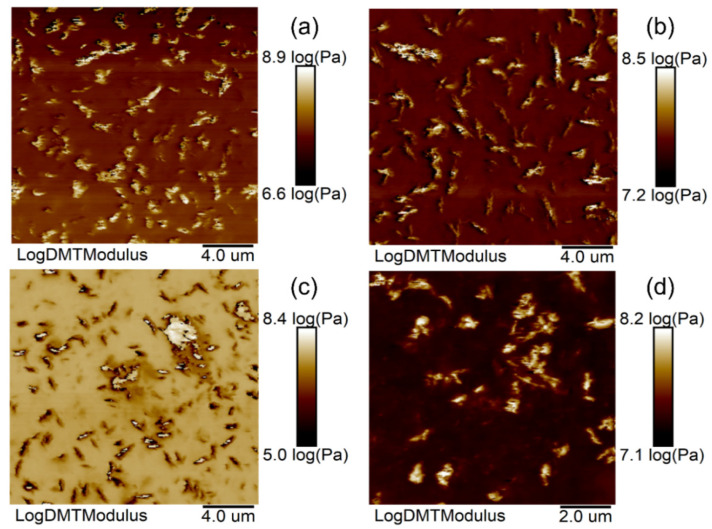
Atomic force microscope logarithmic modulus map of TPU composite doped with different amounts of MXene and CNTs: (**a**) 0.1 wt% MXene/0.2 wt% CNTs, (**b**) 0.25 wt% MXene/0.4 wt% CNTs, (**c**) 0.5 wt% MXene/0.6 wt% CNTs, (d) 1 wt% MXene/0.8 wt% CNTs.

**Table 1 polymers-14-02157-t001:** Dielectric and mechanical properties of TPU composites.

Specimens	Young’s Modulus (MPa)	β (MPa^−1^)	Dielectric Constant at 10^3^ Hz	Dielectric Loss at 10^3^ Hz	Conductivity (S/cm)	*E_b_* (kV/mm)
TPU	4.48 ± 0.06	1.5	6.7	0.047	2.02 × 10^−10^	14.18
M_1_/TPU	2.21 ± 0.03	3.18	7.0	0.411	1.21 × 10^−9^	37.92
C_1_/TPU	2.59 ± 0.02	4.86	12.6	0.441	3.56 × 10^−9^	66.67
CA_1_/TPU	4.45 ± 0.03	2.22	9.9	241.7	1.17 × 10^−6^	10.16
M_1_/C_1_/TPU	1.49 ± 0.03	15.32	22.8	0.476	6.97 × 10^−9^	2.44
M_1_/CA_1_/TPU	8.76 ± 0.05	1.02	9.0	0.086	4.94 × 10^−10^	21.09
M_2_/TPU	2.28 ± 0.05	3.43	7.8	0.240	1.85 × 10^−9^	67.21
C_2_/TPU	2.64 ± 0.04	9.23	24.4	1106.71	1.68 × 10^−5^	1.94
CA_2_/TPU	14.70 ± 0.06	1.21	17.8	1077.23	1.26 × 10^−5^	5.55
M_2_/C_2_/TPU	2.06 ± 0.05	12.67	26.1	0.463	7.75 × 10^−9^	0.397
M_2_/CA_2_/TPU	9.82 ± 0.08	1.08	10.6	0.382	2.60 × 10^−9^	5.04

Note: M_1_(0.25 wt% MXene), M_2_(0.5 wt% MXene), C_1_(0.4 wt% CNTs), C_2_(0.6 wt% CNTs), CA_1_(0.4 wt% CNTs@PDA), CA_2_(0.6 wt% CNTs@PDA).

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
