# Peer review of "Structure and Dielectric Properties of TPU Composite Filled with CNTs@PDA Nanofibers and MXene Nanosheets"

_polymers, 2022, doi:10.3390/polym14112157_

Round 1

Reviewer 1 Report

The manuscript "Structure and dielectric propertises of TPU composite filled with MXene nanosheets and CNTs@PDA nanofibers", is well written and organized. However, few suggestions are put forward.

  1. The Introduction needs improvement.
  2. The environmental conditions during measurement should be mentioned in the manuscript.
  3. How accurate the measurements are? The errors during measurement need to be mentioned.
  4. The Conclusion is absurd and needs improvement.

Author Response

The Comments of the Reviewer #1: The manuscript "Structure and dielectric propertises of TPU composite filled with MXene nanosheets and CNTs@PDA nanofibers", is well written and organized. However, few suggestions are put forward.

1.The Introduction needs improvement.

Re: Thanks for the reviewers comments to improve the quality of our manuscript. According to the reviewer’s suggestion, we have revised the introduction of this manuscript.

2.The environmental conditions during measurement should be mentioned in the manuscript.

Re: Thanks for the reviewers valuable suggestion. According to the reviewer’s suggestion, the environmental conditions we examined were 25 ° C and 30% humidity, which have stated in the manuscript(see Table 1 note).

3. How accurate the measurements are? The errors during measurement need to be mentioned.

Re: The reviewers asked a good question. The test accuracy of mechanical properties were well(see Table 1). The dielectric constant and dielectric loss of the samples were measured twice at different positions, and the test results were almost the same.

4. The Conclusion is absurd and needs improvement.

Re: Thanks for the reviews criticism. Our English writing still needs to improvement. The authors don’t think the conclusion in the manuscript is absurd.

Reviewer 2 Report

This is a nice paper reporting the preparation, structure and dielectric properties of nanocomposites based on thermoplastic polyurethane matrix (TPU) and MXene nanosheets and/or CNTs@PDA nanofibers. The authors noticed that no research was reported on the TPU dielectric composites codoped with MXene and CNTs@PDA particles. At this state the manuscript is not suitable for publication and minor revision is needed, as justified in the following points:

1) The electrical contact is very important for evaluation of dielectric properties of a material. How the authors prepared the electrodes on both sides of the samples?

2) The values of dielectric constant are too accurate. In my opinion, the values should be reconsidered to maximum one decimal place accuracy.

3) Why the numerical values of dielectric constant and dielectric loss (Table 1) are selected at 1 kHz? Following the Figure 9, the frequency dependences of the dielectric constant reveal two different behaviors: 1) a sharp drop of eps’ retrieved at low temperatures, probably assigned to electrode polarization (or MWS) and 2) a moderate decrease of eps’ at high frequencies attributed to the intrinsic dipolar activity of the sample. I believe, the authors should depict the numerical values of eps’ from the moderate decrease of eps’. Since at a frequency of 1 kHz the high polarization process is still present, a higher frequency should be considered.

Author Response

The Comments of the Reviewer #2: This is a nice paper reporting the preparation, structure and dielectric properties of nanocomposites based on thermoplastic polyurethane matrix (TPU) and MXene nanosheets and/or CNTs@PDA nanofibers. The authors noticed that no research was reported on the TPU dielectric composites codoped with MXene and CNTs@PDA particles. At this state the manuscript is not suitable for publication and minor revision is needed, as justified in the following points:

1) The electrical contact is very important for evaluation of dielectric properties of a material. How the authors prepared the electrodes on both sides of the samples?

Re: Thanks for the reviewers valuable comments. We made the electrode liquid on our own. Conductive carbon black, silicone oil, and solvent were used to make the electrode liquid. We take two circular acrylic glass molds with 10 mm holes, clip them to both sides of the dielectric film, and then use an electric spray gun to spray the electrode liquid over both sides of the elastomer film.

2) The values of dielectric constant are too accurate. In my opinion, the values should be reconsidered to maximum one decimal place accuracy.

Re: Thanks for the reviewers careful checks and well suggestion. A broadband dielectric spectrometer instrument was used to deterimine the dielectric constant. According to the reviewer’s suggestion, the value of dielectric constant was revised to maximum one decimal place accuracy.

3) Why the numerical values of dielectric constant and dielectric loss (Table 1) are selected at 1 kHz? Following the Figure 9, the frequency dependences of the dielectric constant reveal two different behaviors: 1) a sharp drop of eps’ retrieved at low temperatures, probably assigned to electrode polarization (or MWS) and 2) a moderate decrease of eps’ at high frequencies attributed to the intrinsic dipolar activity of the sample. I believe, the authors should depict the numerical values of eps’ from the moderate decrease of eps’. Since at a frequency of 1 kHz the high polarization process is still present, a higher frequency should be considered.

Re: Thanks for the reviewers constructive comments. The dielectric constant and dielectric loss measured at 1 kHz are commonly used as the benchmark in the literatures. In general, electron polarization dominates at low frequencies. At high frequencies, directed polarization, or dipole polarization, is the most important factor. At low frequencies, the dielectric constant and dielectric loss of TPU composites decrease dramatically, whereas at high frequencies, dipole polarization tends to stabilize the dielectric constant and dielectric loss.

Higher frequency test values are ignored since the dielectric constant and dielectric loss are only measured in the frequency range of 100 to 108 Hz. We acknowledge the reviewer's recommendations and will compare dielectric performance results at higher frequencies in the future while taking measurements.

Round 2

Reviewer 1 Report

Thank you to the authors for accepting the suggestions and modifying the paper accordingly. Hence, I accept the paper in its current form.

Reviewer 2 Report

The manuscript was correctly revised I accept the publication in the present form.